# Reproducibility Study - SCOUTER: Slot Attention-based Classifier for Explainable Image Recognition

**XXX XXXX**            **XXX XXXX**

**XXX XXXX**            **XXX XXXX**

## Reproducibility Summary

**Scope of Reproducibility**

The experiments presented by Li et al. on SCOUTER were replicated to verify their claims on the properties of the model. For the task of explainable image recognition, the authors state their main claims: (1) SCOUTER produces state-of-the-art performance values on positive and negative explanations. (2) SCOUTER can be trained on different domains with similar accuracy. (3) The area size of the explanatory regions is adjustable by changing the $\lambda$ hyperparameter introduced in the model. (4) The classification accuracy decreases with a larger number of classes.

**Methodology**

A codebase was provided with the paper with the implementation of the model architecture, training and visualization. This was used to replicate the experiments. The calculation of the explanation metrics was partly re-implemented and partly taken from the code base of their respective authors. The experiments supporting the claims were run with adjustments to the number of trials, classes and batch size due to hardware constraints.

**Results**

The replicated experiments could not reproduce most state-of-the-art explanations using SCOUTER with the experimental setup on the evaluation metrics. The experiment using SCOUTER$_+$ with $\lambda = 1$ replicated precision within 3% of the reported value. The accuracy of the model on ImageNet was replicated within 1% of the reported value, however, the accuracy values were not comparably replicated for the CUB-200 dataset. The adjustable area size described by the claim were replicated using SCOUTER$_-$, however, the specific values were not replicated. The experiments could also replicate the trend of decreasing accuracy with respect to an increasing number of classes in the dataset.

**What was easy**

The paper clearly describes the xSlot attention module. Although the code for the evaluation metrics was not provided, references to the papers introducing the metrics were given. The figures and visualizations used in the paper and the appendix were intuitive and useful for understanding their code implementation.

**What was difficult**

The implementation of the evaluation metrics was more difficult than expected since these it was not given and its description does not provide sufficient detail to precisely reproduce their evaluation. Training the models also took longer than anticipated. Although this problem was solved by using remote hardware resources, the cost for training was high and therefore limited the number of experiments that could be performed.

**Communication with original authors**

Authors of the paper were contacted to resolve technical issues: failing to reproduce infidelity. There was no response.

# 1 Introduction

Deep learning models are a family of models characterized by parametric non-linear and hierarchical representation learning functions that encode domain knowledge for a defined task. Due to the large number of parameters and optimization techniques used to find their values, these models are largely viewed as black boxes when used for real world applications. Therefore, a significant disadvantage of using such models is the lack of transparency of a given output when the model is trained to make critical decisions. To alleviate this problem, explainable deep learning models have been proposed which are trained to give an explanation to the decisions made by the model.

The slot-based configurable and transparent classifier model (SCOUTER) [3] was proposed by Li et al. for the task of image recognition which creates visual representations of the explanation of each classification made by their model. This is done by detecting patterns in the input image which propagate positive or negative values to the classification layer. These patterns occur in regions of interest and act as a support for the positive or negative explanations, which are highlighted in the image.

Reproducibility is an important aspect for models in artificial intelligence. It describes how well the performance of a model can be reproduced by an independent group of researchers using the methodology and experiments introduced by the author of the model. This investigation examines the reproducibility of the explainable deep learning model SCOUTER used for image recognition and how well the model generalizes to datasets of other domains.

# 2 Scope of Reproducibility

The authors present sets of experimental results using SCOUTER to support their claims regarding the properties of the model. This investigation will evaluate the reproducibility of the experiments using the experimental setup as described in the paper. Using this method, the following claims will be tested:

- The SCOUTER model can be trained to give state-of-the-art performance for the task of image recognition in terms of precision, IAUC, DUAC, infidelity and sensitivity. This is shown by comparing these performance values with other state-of-the-art models, such as SS-CAM, RISE, I-GOS, IBA and other models.

- The SCOUTER model can be used in different dataset domains with comparable classification accuracy. Therefore, the model is not limited to specific domains of image datasets and can be used in different fields. This is shown by experiments using different datasets such as the CUB-200 and a subset of ImageNet.

- The area size of the regions that contain the supports for the explanation to the output given by SCOUTER can be adjusted by changing the $\lambda$ regularization parameter in the SCOUTER loss equation. This is shown by experiments using different lambda values, which results in a change of area size and small changes in classification accuracy.

- The performance of SCOUTER decreases with an increasing number of classes in the dataset. Therefore, a limitation of the model is that it can only be effectively used for small to medium sized datasets due to difficulty in finding distinct regions of interest over a large number of classes. This is shown by experiments using different number of classes and for the SCOUTER model.

# 3 Methodology

The SCOUTER model architecture, datasets, choice of hyperparameters and experimental setup is described in this section. The implementation of the model and experiments were given by the authors of the paper. This includes the code used for splitting the data, training and visualizations of the results of the SCOUTER model. Since SCOUTER was compared to other state-of-the-art models for explainable image recognition, the code base also includes experiments for other models which were taken from their respective authors. However, the code for these models were not used to test the reproducibility of the SCOUTER results.

## 3.1 Model Architecture

Conventional deep learning models for image recognition consist of a backbone which extracts useful features from an input image and a classifier that uses fully-connected (FC) layers to model a non-linear relation between the features and their correct classification. The SCOUTER model replaces the the latter part of the network. This model consists of the xSlot attention module which calculates the confidence score for each classification [3].

The slot attention mechanism represents a single local region based on the attention over the features derived from the backbone. Therefore, a slot module with multiple attention slots is used to attend over multiple regions of the image.

This is useful when there are different regions of interest that can act supports for a classification. In the xSlot attention module of the SCOUTER model, each slot corresponds to an object class $l$ of the dataset, such that the slot finds the region $S_l$ which is the positive or negative support for the classification $l$ for that image.

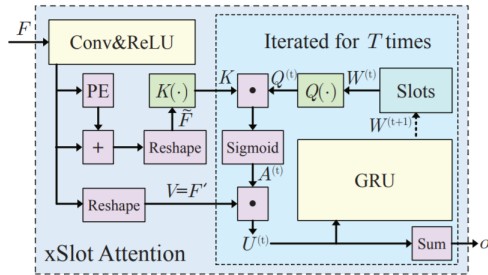

Figure 1: The xSlot attention module in SCOUTER [3]

The backbone network extracts features $F$ which is taken as input by the xSlot attention module. For a given set of features, the module parameters $w_l^{(t)}$ is updated $T$ times, where $w_l^{(t)}$ corresponds to class $l$ after $t$ updates. The initialization of $w_l^{(0)}$ is given as:

$$w_l^0 \sim \mathcal{N}(\mu, diag(\sigma)) \in \mathbb{R}^{1 \times c'}$$

where $\mu$ and $\sigma$ are the mean and variance of a Gaussian distribution, and $c'$ is the size of the weight vector.

The weights for all categories after $t$ updates is denoted by $W^{(t)} \in \mathbb{R}^{n \times c'}$. The updated values for $W^{(t+1)}$ is found using $W^{(t)}$ and $F$. This is shown in Figure 1. $F$ is first propagated through a $1 \times 1$ convolution layer to reduce the number of channels with ReLU activation. The output is flattened such that the output $F' \in \mathbb{R}^{c' \times d}$ (where $d = height \times width$). The position embedding (PE), which encodes the spatial information of each pixel, is appended to $F'$, denoted as $\tilde{F}$. In the next step, two multilayer perceptrons, $Q$ and $K$, are computed which act as the query and key terms in a self-attention mechanism. $Q$ and $K$ consist of three FC layers with ReLU activation with the inputs:

$$Q(W^{(t)}) \in \mathbb{R}^{n \times c'}$$

$$K(\tilde{F}) \in \mathbb{R}^{c' \times d}$$

Self-attention is calculated as:

$$A^{(t)} = \sigma(Q(W^{(t)})K(\tilde{F})) \in (0, 1)^{n \times d}$$

This is used to compute the attention-weighted sum of features:

$$U^{(t)} = A^{(t)} F'^T$$

Lastly, a gate recurrent unit (GRU) is used with $U^{(t)}$ as input and $W^{(t)}$ as the hidden state. The output is the updated $W^{(t+1)}$ term. For each input to the xSlot attention module, $W^{(t)}$ is updated $T = 3$ times. The output score of the xSlot attention module for each class $l$ is given by the sum of the all of the elements in $w_l^{(T)}$. This is formulated as:

$$\texttt{xSlot}(F) = U^{(T)} \mathbf{1}_{c'}$$

where $\mathbf{1}_{c'}$ is a column vector of size $c'$ with all elements having the value of 1. The output is represented by the $o$ term in Figure 1. This equation can be reformulated as $\texttt{xSlot}(F) = A^{(T)} F'^T \mathbf{1}_{c'}$, where the $l$th row of $A^{(T)}$ represents the attention over the spatial map given by $F'^T \mathbf{1}_{c'}$. This corresponds to the support regions of class $l$ which can be visualized by reshaping each row of $A^{(T)}$ into the input image shape. To switch between positive and negative explanations, the model output is modified with hyperparameter $e$ before training:

$$\texttt{xSlot}_e(F) = e \cdot U^{(T)} \mathbf{1}_{c'}$$

where $e \in \{-1, 1\}$. The sign of this hyperparameter corresponds to the positive and negative supports given by the attention values in $A^{(T)}$.

### 3.2 Loss Function

The cross-entropy loss function is used on the softmax of the output of the SCOUTER model. However, this alone causes the model to attend over a large region of the image. Therefore, a regularization term is added:

$$\mathcal{L}_{SCOUTER} = \mathcal{L}_{CE} + \lambda\mathcal{L}_{AREA}$$

$$\mathcal{L}_{AREA} = \mathbf{1}_n^T A^{(T)} \mathbf{1}_d$$

where $\mathcal{L}_{AREA}$ is a sum of all of the elements in $A^{(T)}$. This loss function causes the network to also minimize the area it is attending. The hyperparameter $\lambda$ is used to control the size of that region.

### 3.3 Evaluation Metrics

Some of the evaluation metrics were provided in the authors' code, such as classification accuracy and area size. Precision, IAUC, DAUC, infidelity and sensitivty had to be re-implemented. It was initially not clear how the attention map, which the metrics are calculated from, could be obtained from the given code since it was not explicitly returned by any function. Precision was implemented as described in the paper:

$$p = \frac{\sum_{i \in D} a_i^l}{\sum_{i \in I} a_i^l}$$

where $I$ is the set of all input pixels and $D$ is the set of pixels within the bounding box. This equation of precision seemed open to interpretation, since some values of the bounding boxes were outside the bounds of the images. This is because the original images are scaled down to a size of $260 \times 260$ to be compatible with the model and that the attention $A^{(T)}$ is scaled up to the size of the input images using bilinear interpolation. This could either be interpreted as resizing $A^{(T)}$ to the size of the original image or that of the input to the model. In this evaluation, the bounding box labels were scaled down from their original size to $260 \times 260$ and the size of the attention map was scaled up from $9 \times 9$ to the $260 \times 260$ to match the sizes of the bounding boxes to the attention map.

The insertion area under curve (IAUC) and deletion area under curve (DAUC) measure the changes in the accuracy of the model when inserting and deleting pixels from the attention map in the input image, respectively. The code implementation of the IAUC and DAUC was provided by Petsiuk et al. in their paper[4]. Their functions evaluate multiple explanation models. This was modified to evaluate a single explanation model for SCOUTER.

The equations for sensitivity and infidelity were given in the paper[6] by Yeh et al. and the implementation was provided online.

$$SENS_{MAX} = (\Phi, x) = \max_{||y-x|| \leq r} ||\Phi(y) - \Phi(x)||$$

where $\Phi$ is the SCOUTER model, $x$ is the image input, $r$ is the input neighborhood radius and $y$ is the perturbed input [2]. This quantifies the change in the explanation when there is a small perturbation in the input. The code was adjusted for to be compatible with SCOUTER explanations. There were two possible methods for implementing sensitivity in the case where the perturbation changes the prediction class. The attention of the original class could be used to calculate sensitivity or the attention of the perturbed classification. Since this was not specified, the second option was chosen.

There were implementation or computational errors with the code for calculating infidelity. Therefore, this metric was not used in this study. The authors of the SCOUTER model were contacted regarding the evaluation metrics, however there was no clarifications were given.

### 3.4 Datasets

Two datasets were used in the SCOUTER experiments. The dimensionality of the images were $260 \times 260$. The pixel values of ImageNet follow a Gaussian distribution with a mean of $(0.485, 0.456, 0.406)$ and standard deviation of $(0.229, 0.224, 0.225)$ for the RGB values. The images in this subset and of the other datasets are normalized to this Gaussian distribution.

The ImageNet dataset consists of a large set of images of a variety of object classes [1]. A subset of ImageNet containing the first 100 classes based on class ID is used for all experiments in the paper.

CUB-200 contains 200 classes of bird species [5]. This is a more difficult dataset to solve since there is less variation between the classes than in the ImagNet subset.

### 3.5 Experimental Setup and Hyperparameters

A large number of training runs is required to replicate all of the results of the original paper. Due to time constraints, a subset of these experiments were done which may sufficiently represent the trends and properties proposed by the authors. Most of the settings and hyperparameters were the same as given in the paper and the provided code.

Settings of some of the experiments were changed in the replication experiments. SCOUTER was trained using ImageNet to generate the state-of-the-art explanations. ImageNet was also used to train with $\lambda$ values of $\{1, 3, 10\}$ to verify changes in area sizes of the explanatory regions. CUB-200 was used to train SCOUTER to test for its classification accuracy with different number of classes instead of ImageNet. This allowed for more training runs due to a shorter training time. The numbers of classes that were tested are $n = \{25, 50, 75, 100\}$ and the ResNeSt50 backbone was used instead of ResNeSt26. The default batch size for all experiments was 70, however this was changed to 16 for the CUB-200 dataset. This was done to avoid memory errors caused by the large batch size and computational limitations.

The code implementations of the reproducibility experiments can be found in the Github repository of this project: https://anonymous.4open.science/r/FACT-2022-B40D/README.md.

### 3.6 Computational Resources

Initially, the experiments were run on the XXXX server of XXXX using NVIDIA Titan RTX, 24 GB GDDR6. However, the runtime was very high with the longest experiment running for 10 hours using the ImageNet subset. The experiments exhausted the allowance of GPU hours on the XXXX server. For further experiments, the Google Cloud Platform was used with NVIDIA Tesla V100. The runtime with this setup for the longest experiment was 4.5 hours.

## 4 Results

### 4.1 Evaluation of Explanation Results

The explanation results are evaluated with a number of metrics as shown in Table 1. The original performance values support the claim that SCOUTER produces state-of-the-art positive and negative explanations. The results of the reproducibility experiments show that few of the original values of precision, IAUC, DUAC and sensitivity could be replicated. The precision of SCOUTER$_+$ with $\lambda = 1$ was closely reproduced. However, the other precision and sensitivity values were not replicated. The IAUC and DUAC values were incorrectly calculated since there are negative values and not normalized between 0 and 1.

Table 1 also shows the area size of the explanatory regions for experiments using $\lambda$ values of 1, 3 and 10. The original area size values are given to support the claim that a higher $\lambda$ value causes the area size to decrease and the area size can be accordingly adjusted for smaller and more precise explanations. This trend is observed in the reproduced area sizes for SCOUTER$_-$ but not for SCOUTER$_+$. The values of the area sizes in general are also different, except for SCOUTER$_+$ with $\lambda = 1$.

| | Model | Area Size | Precision | IAUC | DAUC | Sensitvity |
|---|---|---|---|---|---|---|
| Original | SCOUTER$_+$ ($\lambda = 1$) | 0.1561 | 0.8493 | 0.7512 | 0.1753 | 0.0796 |
| | SCOUTER$_+$ ($\lambda = 3$) | 0.0723 | 0.8488 | 0.7650 | 0.1423 | 0.0608 |
| | SCOUTER$_+$ ($\lambda = 10$) | 0.0476 | 0.9257 | 0.7647 | 0.2713 | 0.1150 |
| Reproduced | SCOUTER$_+$ ($\lambda = 1$) | 0.149 | **0.820** | -5.341 | -0.109 | 0.258 |
| | SCOUTER$_+$ ($\lambda = 3$) | 0.158 | 0.777 | -4.225 | -0.649 | 0.333 |
| | SCOUTER$_+$ ($\lambda = 10$) | 0.157 | 0.814 | -4.283 | -0.117 | 0.268 |
| Original | SCOUTER$_-$ ($\lambda = 1$) | 0.0643 | 0.8238 | 0.7343 | 0.1969 | 0.0567 |
| | SCOUTER$_-$ ($\lambda = 3$) | 0.0545 | 0.8937 | 0.6958 | 0.4286 | 0.1497 |
| | SCOUTER$_-$ ($\lambda = 10$) | 0.0217 | 0.8101 | 0.6730 | 0.7333 | 0.1895 |
| Reproduced | SCOUTER$_-$ ($\lambda = 1$) | **0.021** | 0.778 | -5.490 | -0.100 | 0.350 |
| | SCOUTER$_-$ ($\lambda = 3$) | **0.012** | 0.798 | -4.230 | -0.118 | 0.486 |
| | SCOUTER$_-$ ($\lambda = 10$) | **0.005** | **0.929** | -4.262 | -0.131 | 0.277 |

Table 1: Evaluation Measures of Positive and Negative Explanation Results with ResNest26 and $n = 100$ for different values of $\lambda$.

## 4.2 Classification Accuracy on Different Datasets

The positive and negative SCOUTER models were evaluated, as well as the network with a FC classifier as baseline, on a 100-class subset of ImageNet and CUB-200. The original results support the claim that SCOUTER can be used across domains and also in more specialized domains, like CUB-200 with comparable classification accuracy. The accuracy values for ImageNet was successfully reproduced as shown in Table 2. However, there was a significant drop in accuracy when the experiment was replicated for CUB-200 using SCOUTER.

|  | Model | ImageNet | CUB-200 |
|---|---|---|---|
| Original | FC | 0.8080 | 0.7538 |
|  | SCOUTER$_+$ | 0.7991 | 0.7362 |
|  | SCOUTER$_-$ | 0.7946 | 0.7490 |
| Reproduced | FC | 0.804 | **0.781** |
|  | SCOUTER$_+$ | 0.794 | 0.474 |
|  | SCOUTER$_-$ | 0.795 | 0.646 |

Table 2: Classification accuracy of SCOUTER trained on different datasets using ResNeSt26 with $n = 100$ and $\lambda = 10$

## 4.3 Effect of Hyperparameter $\lambda$ on Area Size

Figures 2 and 3 show the visual representation of the explanatory regions produced by SCOUTER$_+$ and SCOUTER$_-$ respectively. The SCOUTER$_+$ model does not support the claim that $\lambda$ can be used to adjust the area size, since the area size does not necessarily decrease for increasing values of $\lambda$, as shown in Figure 2. However, the SCOUTER$_-$ model does support this trend, as shown in Figure 3.

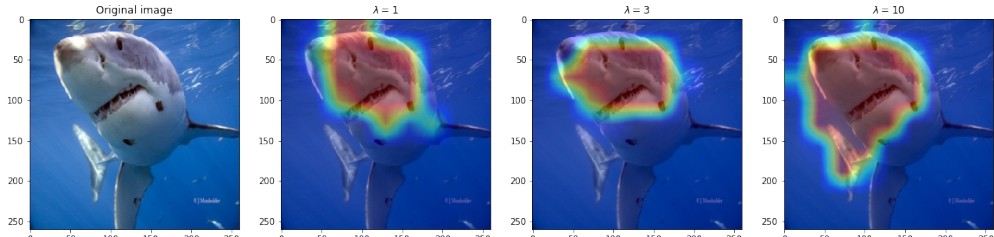

Figure 2: Positive SCOUTER area size as $\lambda$ increases example for a shark image

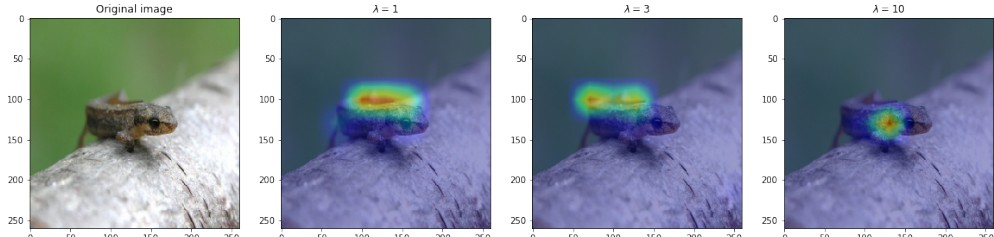

Figure 3: Negative SCOUTER area size as $\lambda$ increases example for a frog image

## 4.4 Effect of Number of Classes on Classification Accuracy

SCOUTER was trained with the CUB-200 dataset to verify the claim that its classification accuracy decreases with an increasing number of classes. This claim was successfully reproduced as shown in Figure 4 that accuracy generally decreases. This is a result of unstable training which can be seen in Figures 5, 6 and 7 in the appendix.

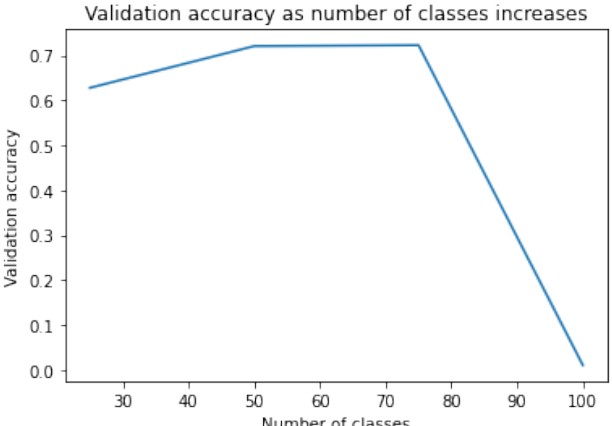

Figure 4: Classification accuracy on CUB-200 with SCOUTER$_+$ and $\lambda = 10$ in relation to the number of classes in the training and validation set.

## 5 Discussion

This study attempted to replicate the results of "SCOUTER: Slot Attention-based Classifier for Explainable Image Recognition" that support the claims about the properties of the model. The results show that the state-of-the-art performance of SCOUTER on positive and negative explanations could generally not be replicated with the given experimental setup. There was a significant drop in accuracy when the model was trained using CUB-200 dataset. This setup does not support the claim that SCOUTER can be used with other datasets with similar accuracy. The results of area size in relation to $\lambda$ support the claim that area size can be adjusted for negative SCOUTER explanations, but it is not clear for positive SCOUTER explanations. The $\mathcal{L}_{AREA}$ loss curve of SCOUTER$_+$ during training shows that it drops to zero. Therefore, the $\lambda$ value does not affect area size in that case. This may be caused by the ReLU function applied to $\mathcal{L}_{AREA}$ in the code. This operation is not described in the original paper. The last set of results show a trend of decreasing accuracy with an increasing number of classes which supports that claim.

Generally, the results of the evaluation metrics were different from that of the original paper. The IAUC and DAUC scores were incorrect and the sensitivity is consistently higher than that of the original paper. This is because of the specific implementation of the metrics. Although the provided code from the referenced papers was used, it was modified in order to be compatible with the attention map. Certain parameters of the metrics were adjusted. This includes step-size in IAUC and DAUC and SEN$_r$ in sensitivity which controls the amount of noise. A parameter search could be done to possibly match the experimental setting and improve the metric results.

The implementation of precision may be different from that of the original paper. Since the precision scores are still high, few of them closely match the original precision, and for the negative SCOUTER with $\lambda = 10$, it exceeds the reported value. This strengthens the claim that the model performs state of the art on explainability in some of the metrics.

### 5.1 What was easy

The authors provided a code base and clearly describe the xSlot attention module in the paper. Although, the code for the evaluation metrics were not provided, references to the papers introducing the metrics were given. The figures and visualizations used in the paper were intuitive and useful for understanding their code implementation.

### 5.2 What was difficult

The implementation of the evaluation metrics was more difficult than expected. It often involved guesswork since the implementation of the metrics was not given and its description does not provide sufficient detail to precisely reproduce their evaluation. It was unclear how the attention map should be used for the metrics. An equation is given for precision, however it is also unclear whether the attention map should be resized to the input image of the model or the original image. Therefore, a considerable amount of time was spent on writing a working implementation of parsing the bounding boxes from the dataset and calculating each of the metrics. The paper also states that the IAUC and DAUC use heat maps and provide the code for generating RGB heat maps. However, using the heat map did not improve

results over using the original single channel attention map with floating point values between 0 and 1. It was also not clear which of these values (floating point or integers between 0 and 255) were used for sensitivity and infidelity.

Training the models also took longer than anticipated, since the code was not easy to run locally. This was because it required multiple training runs on a single model while making heavy use of the checkpoint feature to prevent "out of memory" error. This was also done with lower batch sizes and number of workers. Only one local machine would run the training, therefore fewer experiments could be run locally. After using up the allowance of GPU hours on XXXX, two Google Cloud accounts were used with EUR 300.00 of free credits. All further experiments could be run on the GPU provided in Google Cloud.

### 5.3 Communication with original authors

Due to the difficulties of reproducing the evaluation metrics, two primary authors of the paper (Liangzhi Li and Manisha Verma) were contacted with the aim to get further clarification and assistance on this issue.

In particular, the infidelity metric was very difficult to reproduce. The original authors of the metric [6] provide scripts, but applying them directly to the SCOUTER code base introduced memory errors. This caused both the physical computers and instances to shut down and reboot.

As of the 4th of February 2022, no response was received from either authors.

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

# Appendix

## 5.4 SCOUTER Accuracy Curves using CUB-200

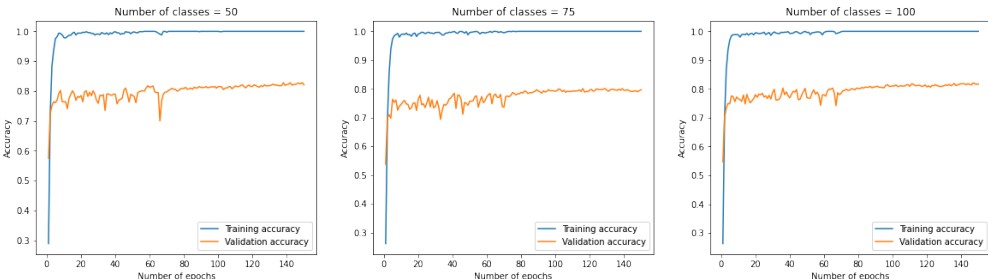

Figure 5: Training and validation accuracy curves of FC classifier for different number of classes using CUB-200 and $\lambda = 10$

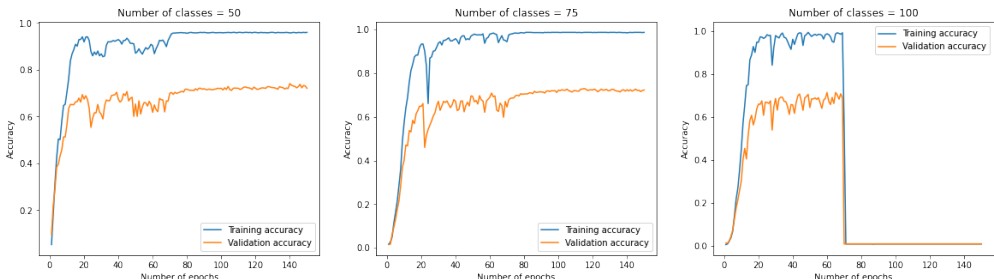

Figure 6: Training and validation accuracy curves of SCOUTER$_+$ for different number of classes using CUB-200 and $\lambda = 10$

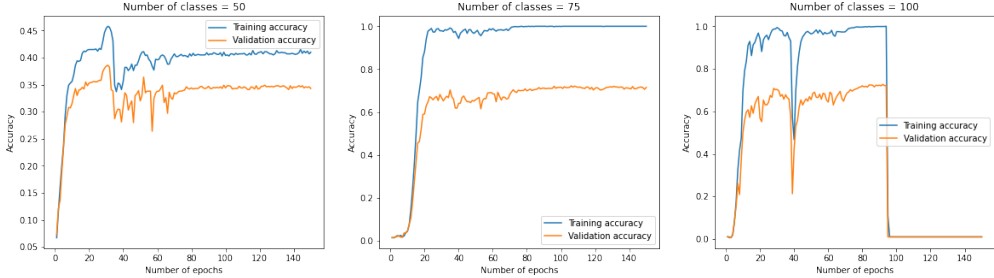

Figure 7: Training and validation accuracy curves of SCOUTER$_-$ for different number of classes using CUB-200 with $\lambda = 10$

