# OpenReview forum: "Reproducibility Study - SCOUTER: Slot Attention-based Classifier for Explainable Image Recognition"
_ML_Reproducibility_Challenge/2021/Fall — Reject_

### Official Review · Reviewer_SRoe · 2022-03-08
**Some experiments could be reproduced. Implementation details and lack of additional works seem to impair reproducibility claims.**

**Rating:** 4
**Confidence:** 4

**Review:**

This work aims to reproduce the key results of ‘SCOUTER: Slot Attention-based Classifier for Explainable Image Recognition’ by Li et al.

The authors very clearly state which experiments are done and what results are tried to be reproduced. For those experiments the original codebase of Li et al. was used, only the missing evaluation metrics Precision, IAUC, DAUC, infidelity, and sensitivity had to be implemented by the authors. The documentation of the written code seems good enough. Multiple issues were faced while implementing these metrics, where the main obstacle seem to be how to scale the attention map. The authors opt for scaling down to 260x260 and scale up the attention from 9x9 pixels to 260x260 pixels to match the sizes of the bounding boxes of the attention map without giving further justification. Upon request trying to resolve implementation details the authors did not receive an answer from Li et al. Though, as the metrics play a crucial role in the evaluation of the reproducibility, the authors could have done more rigorous experimentation trying out different options. IAUC and DAUC were implemented incorrectly, as mentioned by the authors. Infidelity could not be reproduced due to “implementation or computational errors”.

Due to time constraints, the authors decide to change the parameters for some experiments. From their description, it is not clear whether their experiments are different from the original ones or whether only a subset of the experiments (using the same parameters) was carried out. Section 3.5 seems generally unclear. No supplementary experiments or additional hyperparameter searches were carried out. Also, no ablation studies were done to verify/challenge the original experiments.

The discussion of the results seems clear. No recommendations to the original authors to improve reproducibility are made.

I feel that ruling out ambiguities of the implementation details did not receive enough attention. Additionally, some parameters have been altered non-systematically compared to the original work. Even though some of the results could be reproduced, I have my doubts whether this paper can be conclusive about the reproducibility of the original work by Li et al.  Additional experiments, analysis or ablation studies could have supported their reproducibility claims.


Other Remarks:

- Putting the file into reviewing mode would have eased the review process as no line numbers are provided in the pdf.
- Where the formula for self-attention is introduced “n” is undefined, Section 3.1.
- It would have been nice to see the decrease in performance to the original paper as well, Figure 4

Grammatical issues:

more difficult than expected since these it was not given -> more difficult than expected since it was not given, page 1

model replaces the the latter part -> model replaces the latter part, page 2

---

### Official Review · Reviewer_UsEP · 2022-03-08
**Well written approached report**

**Rating:** 8
**Confidence:** 4

**Review:**

Authors provided very thorough detail on their reproducibility efforts. Their report has been well written and concepts have been well articulated. They managed to reproduce the results of the original papers with a very good precision, albeit with some deviations.
However, they tried to contact the authors to get some further insights on certain functions, e.g. infidelity metric, but they did not receive a response.

Considering all the above, authors did a great job in reproducing the original paper to a good precision level.

---

### Meta-Review · Program_Chairs · 2022-04-07

**Recommendation:** Reject
**Confidence:** 4

**Metareview:**

While the reproducibility itself is well carried out, there a lot of missing pieces of information regarding the implementation itself and the choices that were made regarding parameters, etc. For example, if this is due to computational resources constraints, then this should be clearly indicated; if there are other reasons, those should be communicated as well.

---

### Decision · Program_Chairs · 2022-04-09

Reject